# Proactive Deepfake Detection via Training-Free Landmark Perceptual Watermarks

Submission Id: 682*

## ABSTRACT

The Deepfake face manipulation technique has garnered significant public attention due to its impacts on both enhancing human experiences and posing security and privacy threats. Despite numerous passive Deepfake detection algorithms that have been attempted to thwart malicious Deepfake attacks, they mostly struggle with the generalizability challenge when confronted with hyper-realistic synthetic facial images contemporarily. To tackle the problem, this paper proposes a proactive Deepfake detection approach by introducing a novel training-free **la**nd**m**ark **p**erceptual water**mark**, *LampMark* for short. Firstly, we analyze the structure-sensitive characteristics of Deepfake manipulations and devise a secure and confidential transformation pipeline from the structural representations, i.e. facial landmarks, to binary landmark perceptual watermarks. Subsequently, we present an end-to-end watermarking framework that robustly and imperceptibly embeds and extracts watermarks concerning the images to be protected. Relying on promising watermark recovery accuracies, Deepfake detection is accomplished by assessing the consistency between the content-matched landmark perceptual watermark and the robustly recovered watermark of the suspect Deepfake image. Experimental results demonstrate the superior performance of our approach in watermark recovery and Deepfake detection compared to state-of-the-art methods across in-dataset, cross-dataset, and cross-manipulation scenarios.

## CCS CONCEPTS

• **Security and privacy** → *Digital rights management*; **Social aspects of security and privacy**; • **Computing methodologies** → **Computer vision problems**.

## KEYWORDS

Deepfake detection, landmark perceptual watermark, digital forensics, robust watermarking.

**ACM Reference Format:**
Anonymous Author(s). 2024. Proactive Deepfake Detection via Training-Free Landmark Perceptual Watermarks. In *Proceedings of the 32nd ACM International Conference on Multimedia (MM '24)*. ACM, New York, NY, USA, 10 pages. https://doi.org/XXXXXXX.XXXXXXX

## 1 INTRODUCTION

Deepfake, a deep neural network based facial manipulation technique, has yielded substantial effects on society from both positive and negative perspectives [40]. While being adopted in industries such as filmmaking and education for benign utilization, Deepfake attacks have severely jeopardized the privacy and security of human beings. To satisfy the demand of preventing the current and potential risks accordingly, abundant attempts for Deepfake detection have been conducted in the research domain.

Most existing Deepfake detection approaches fall in the category of passive detection. In essence, the algorithms are designed to distinguish between real and fake facial images after Deepfake has manipulated the original real images. While fake images can mostly be detected by tracing the synthetic artifacts within image feature domains in early stages, most methods have struggled with bottlenecks when facing hyper-realistic Deepfake contents since no obvious manipulation trace can be explicitly or implicitly located. This is also reflected in the unsatisfactory and fluctuating generalizability of the passive detectors on unseen manipulations and datasets.

Recently, the concept of proactive defense has been raised such that invisible signals are inserted into benign images in advance of potential manipulations and falsifications can be addressed regarding their existence. This concept encompasses two topics, distorting and watermarking. While the former [14, 35] directly adds learned noises into images to disable Deepfake manipulations regardless of malicious and benign purposes, the latter protects the images in relatively subtle manners. On the one hand, semi-fragile watermarks [26, 50] are vulnerable to Deepfake manipulations, and the detection is executed based on the absence of watermarks. On the other hand, robust watermarks [38, 51] are used for attribution, detection, and source tracing purposes, owing to their distinct characteristics and semantics. Despite preliminary attempts in the domain, considerable research gaps persist, particularly concerning the robustness and generalizability of proactive watermarks.

To mitigate these problems, we delve into the common behaviors of Deepfake manipulations in the widely known two categories, face swapping and face reenactment. Face swapping modifies facial identities and maintains other facial attributes unchanged. Conversely, face reenactment reconstructs facial expressions and head poses but preserves the facial identities. Although they each follow a unique protocol in the synthetic pipeline, the facial structures are generally modified regardless of identities, expressions, and head poses. In this paper, we exploit the facial landmarks to represent the structural information of facial images and construct landmark perceptual watermarks accordingly for proactive Deepfake detection. As Figure 1 depicts, facial landmarks of the images after benign image processing operations and Deepfake manipulations are extracted and compared with those of the original raw images. In


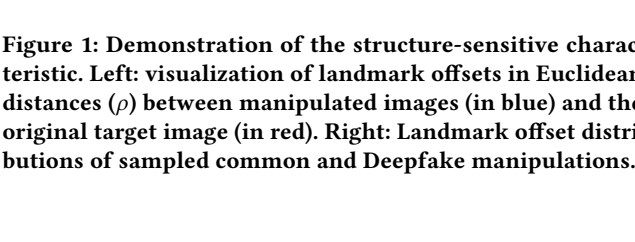

**Figure 1: Demonstration of the structure-sensitive characteristic. Left: visualization of landmark offsets in Euclidean distances ($\rho$) between manipulated images (in blue) and the original target image (in red). Right: Landmark offset distributions of sampled common and Deepfake manipulations.**

the left sub-figure, obvious offsets can be observed regarding face swapping (SimSwap [6]) and face reenactment (StarGAN [7]) manipulations. In contrast, only relatively imperceptible offsets are caused by the benign manipulation (GaussianNoise). Scientifically, 10K images are randomly sampled from the CelebA-HQ [18] dataset to demonstrate a prevalent pattern in landmark offsets caused by image manipulations. In particular, four benign manipulations (Jpeg, GaussianNoise, GaussianBlur, MedianBlur), two face swapping manipulations (SimSwap and InfoSwap [9]), and two face reenactment manipulations (StarGAN and StyleMask [3]) are adopted to produce images on the 10K ones. As exhibited in the right sub-figure of Figure 1, a pellucid gap is observed between landmark offset distributions in Euclidean distances caused by benign and Deepfake manipulations. In a nutshell, structures of facial landmarks are unavoidably modified upon Deepfake manipulations, while benign image operations generally maintain the original image content.

In this study, to proactively protect images against malicious Deepfake manipulations, we analyze the structural consistency of facial landmarks before and after image manipulations and propose a **la**nd**m**ark **p**erceptual water**mark**ing framework, namely, *LampMark*. First, we introduce a training-free watermark construction pipeline that projects the coordinates of facial landmarks to binary watermarks with fixed lengths while preserving the characteristics and distributions as displayed in Figure 1. Then, we devise a cellular automaton encryption system to securely encrypt the watermarks with unpredictable and complex manners, guaranteeing strong watermark confidentiality. Thereafter, we train an end-to-end watermarking framework that robustly embeds and recovers watermarks against benign image processing operations and Deepfake manipulations. For a watermarked image that is suspected to be fake, Deepfake detection is achieved by analyzing the similarity between the recovered watermark and the landmark perceptual watermark with respect to the suspect image. Extensive experiments demonstrate outstanding watermark robustness with average bit-wise watermark recovery accuracies of 91.83% and 91.86% on CelebA-HQ at 128 and 256 resolutions, respectively. Furthermore, obtaining 98.39% and 98.55% AUC scores on detecting a mixed set of seven Deepfake manipulations demonstrates the

state-of-the-art performance of our approach. The contributions of this work can be summarized as follows:

- We exploit the structure-sensitive characteristic of facial landmarks regarding Deepfake manipulations and devise novel training-free landmark perceptual watermarks with confidentiality to defend against Deepfake proactively.
- We propose a framework that robustly inserts and extracts the landmark perceptual watermarks into and from facial images. To the best of our knowledge, we are the first to simultaneously detect face swapping and face reenactment Deepfake manipulations with a single robust watermark.
- Extensive experiments under in-dataset, cross-dataset, and cross-manipulation settings demonstrate the promising watermark recovery and Deepfake detection performance of our method, outperforming the state-of-the-art algorithms.

## 2 RELATED WORK

### 2.1 Deepfake Generation

Ever since the first occurrence raised on Reddit[1] in 2017, the term 'Deepfake' has attracted abundant public attention. Throughout the evolution of Deepfake technology, the generative algorithms have been broadly classified into two categories: face swapping and face reenactment. Face swapping [6, 9, 19, 20, 23, 27, 29, 39, 47] aims to swap the facial identity from a source image onto the target one while preserving the remaining image semantics. On the contrary, face reenactment [2, 3, 7, 11, 45, 49, 52] transfers the facial attributes, including expressions and poses, from a source image onto the target one but maintains the original facial identity. Both categories reconstruct the target image with the desired content modifications that mostly happen on the facial structures. Recently, unlike the early ones that exhibit obvious synthetic traces, Deepfake algorithms employing generative adversarial networks [10] (GANs) have shown promising and efficient performance in producing hyper-realistic synthetic content. These advanced synthetic models have achieved seamless generations without visual artifacts by incorporating modules that implicitly update the underlying features and smoothly enforce changes within the facial areas. While benefiting the industry in several aspects, this has brought severe challenges to the domain of multimedia forensics, especially to the passive Deepfake detection algorithms.

### 2.2 Proactive Deepfake Detection

While the passive Deepfake detection algorithms progressively advance the steps to address the underlying synthetic artifacts, experiencing the evolution of CNN based approaches [1, 55], CNN backbone based approaches [21, 28, 31, 37], integrated architectures [24, 36, 53], dataset enrichment strategies [5, 32, 33, 54], and deep analysis on implicit feature domains [4, 8, 22], they mainly lack generalization ability when encountering newly occurred Deepfake models and datasets. Moreover, the high-quality synthetic outputs of recent generative algorithms further aggravate the challenges in distinguishing the fakes.

Recently, instead of passively engaging in the expensive competition, several studies have been undertaken to proactively defend

[1]https://www.reddit.com/

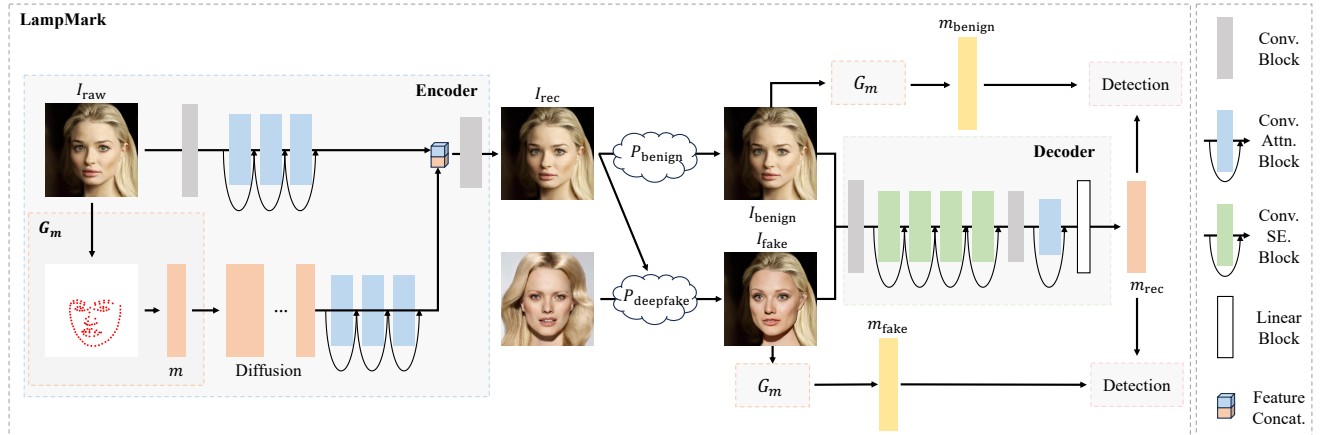

**Figure 2: Overall framework of the proposed method. The landmark perceptual watermarks are produced via pipeline $G_m$ and encoded into raw images. After benign and Deepfake manipulations, the watermarked images are passed to the decoder for watermark recovery. By comparing the landmark perceptual watermarks of the manipulated images with the recovered watermarks, Deepfake detection is accomplished.**

against Deepfake and preemptively protect the raw images in advance. In 2020, Ruiz, Bargal, and Sclarof [30] raised the idea of 'disrupting deepfakes' with a gradient-based method that adds an invisible perturbation $\eta$ to the image $x$ following $\tilde{x} = x + \eta$ such that the resulting image $\tilde{x}$ is able to disrupt the generation of Deepfake. Thereafter, several follow-up studies step further. Huang *et al.* [16] designed a two-stage training framework that superposes perturbations onto images to nullify face attribute editing and reenactment manipulations. While later approaches [14, 35, 41, 57] progressively advance the disruption performance, they unfortunately disable the benign utilization of Deepfake, and the visual qualities of images are unavoidably affected due to directly added perturbations. To resolve the above issues, subsequent attempts are made via watermarking. Yu *et al.* [51] trained a framework that inserts model-specific fingerprints for the attribution purpose. FaceGuard [50] and Face-Signs [26] determine the falsification based on the existence of the embedded semi-fragile watermarks. ARWGAN [15] contains an attention-guided robust watermarking framework that resists GAN model attacks. Wu, Liao, and Ou [46] proposed SepMark, a separable watermarking framework that incorporates semi-fragile and robust watermarks, to perform detection and source tracing via analyzing each watermark. Wang *et al.* [38] fulfilled the entire pipeline of Deepfake face swapping detection by training a robust watermarking framework with watermarks containing identity semantics.

## 3 METHODOLOGY

### 3.1 Problem Formulation

Contrary to the passive detection methods which analyze explicit and implicit artifacts to identify Deepfake images, proactive defense approaches focus on safeguarding the original images before potential synthetic manipulations happen. This is achieved by inserting visually imperceptible information within the image contents. In this study, we introduce a training-free pipeline, denoted as $G_m$,

designed to transform facial landmarks of facial images into landmark perceptual watermarks of fixed lengths. Subsequently, we devise an auto-encoder architecture to robustly embed and recover the watermarks into and from the images, respectively. Finally, we address falsifications by comparing the recovered watermarks with the landmark perceptual watermarks of the manipulated images that are watermark-protected.

In practice, a real image $I_{\text{raw}}$ is embedded with the corresponding landmark perceptual watermark $m_{\text{raw}}$, derived following the pipeline $G_m$, to provide proactive protection, and the watermarked image $I_{\text{rec}}$ is then reconstructed. When a Deepfake manipulation occurs on $I_{\text{rec}}$, generating the synthetic image $I_{\text{fake}}$, two watermarks are obtained accordingly. Firstly, the embedded watermark can be recovered as $m_{\text{rec}}$, faithfully similar to $m_{\text{raw}}$ due to its robustness. Meanwhile, the landmark perceptual watermark $m_{\text{fake}}$ can be obtained via $G_m$ based on the image content of $I_{\text{fake}}$. The bit-wise matching rate between $m_{\text{rec}}$ and $m_{\text{fake}}$ is expected to be relatively low since they are produced based on different images with unique facial landmarks. Additionally, $I_{\text{rec}}$ typically undergoes visual quality degradation (e.g., compression, noising, and blurring) upon uploading and spreading on the internet, becoming the processed image $I_{\text{benign}}$. Consequently, the matching rate between $m_{\text{rec}}$ and $m_{\text{benign}}$ is expected to be high, as their visual contents are generally similar. Ultimately, these matching rate values can assist in determining falsifications.

### 3.2 Landmark Perceptual Watermarks

Motivated by the structural variations in facial images caused by Deepfake manipulations and the structure-sensitive characteristic regarding facial landmarks as demonstrated in Figure 1, we construct landmark perceptual watermarks tailored to this purpose. The watermarks are crafted to differentiate Deepfake-manipulated facial images by verifying the watermark consistency in a secure and confidential manner. Generally, we aim to fit an optimum mapping $\mathcal{H}(\cdot) : L \rightarrow M$ from facial landmarks $L$ to watermarks $M$ that

strictly possesses three characteristics: *Discrimination*, *Confidentiality*, and *Robustness*.

*3.2.1 Discrimination.* Considering the limited capacity in concealing watermarks seamlessly in images, for facial landmarks with a fixed number of points, we propose to reduce the feature dimensions of landmarks while preserving the integrity of the original distribution as summarized in Figure 1. Specifically, principle component analysis (PCA) is employed to transform the landmark points with dimension $(d_{lm}, 2)$ to vector features with length $l$. Upon choosing a data corpus with sufficient quantity and diversity, the landmark points are first flattened to vectors of length $2d_{lm}$, denoted as $E_{lm}$ for the corpus, following the order of $\{x_0, y_0, x_1, y_1, ..., x_{d_{lm}-1}, y_{d_{lm}-1}\}$ such that $x_i$ and $y_i$ are the coordinates. Then, following the algorithm of PCA, the covariance matrix of $E_{lm}$ is computed as

$$\text{Cov}(E_{lm}) = \frac{1}{\text{len}(E_{lm})}(E_{lm} - \mu_{E_{lm}})^{\text{T}}(E_{lm} - \mu_{E_{lm}}), \qquad (1)$$

where $\mu_{E_{lm}}$ refers to the mean vector of $E_{lm}$. Thereafter, the corresponding eigenvectors $\mathbf{v}$ and eigenvalues $\lambda$ are obtained by solving the eigendecomposition equation

$$\text{Cov}(E_{lm})\mathbf{v} = \lambda\mathbf{v}. \qquad (2)$$

The eigenvectors are sorted with respect to the eigenvalues in descending order, representing the principal components of $E_{lm}$. Thenceforth, we preserve the top $l$ eigenvectors to form the projection matrix $\mathbf{W}$ and the transformed vectors with length $l$ are derived following

$$E_{\text{trans}} = E_{lm} \cdot \mathbf{W}. \qquad (3)$$

Since the eigenvectors with the best $l$ values are maintained, features with the least importance are eliminated during dimension reduction, preserving the uttermost information of features and the original distribution.

Finally, to construct watermarks with binary values, we scale the $l$ features of each vector in $E_{\text{trans}}$ to the range of $[0, 1]$. In detail, we normalize $E_{\text{trans}}$ with respect to the upper and lower bounds of all the vector features,

$$E_{\text{norm}} = \frac{E_{\text{trans}}[:, i] - \min(E_{\text{trans}}[:, i])}{\max(E_{\text{trans}}[:, i]) - \min(E_{\text{trans}}[:, i])}, \qquad (4)$$

for feature values at all $l$ indices where $0 <= i < l$. Lastly, setting the threshold at 0.5 enforces all vectors in $E_{\text{norm}}$ contain only binary values, denoted as $E_{\text{bin}}$.

*3.2.2 Confidentiality.* By having access to the pipeline of generating the landmark perceptual watermarks, an attacker may intentionally replace the embedded watermark with a content-matched one so that Deepfake detection is disabled. To ensure the confidentiality of the watermarks, in this study, we leverage the concept of cellular automata [34], which refers to a mathematical modeling paradigm for complex systems, and design a cellular automaton [42, 43] encryption system with a specific transform rule to provide random and chaotic behaviors for the watermarks. Particularly, regarding a binary encryption key with length $l$ in the cellular automaton, the state of each bit at the next time step is determined by the transform rule and the neighboring bits, such that

$$s_i^{t+1} = R(s_{i-1}^t, s_i^t, s_{i+1}^t), \qquad (5)$$

where $s_i^{t+1}$ denotes the state at bit index $i$ for time step $t+1$ following the transform rule $R$.

In this study, Rule 30 [44] is applied and the key set $K$ with an initial key $k_0$ for watermark encryption is derived following

$$s_i^{t+1} = \begin{cases} s_{l-1}^t \oplus (s_0^t \vee s_1^t), & \text{for } i = 0, \\ s_{i-1}^t \oplus (s_i^t \vee s_{i+1}^t), & \text{for } 0 < i < l - 1, \\ s_{l-2}^t \oplus (s_l^t \vee s_0^t), & \text{for } i = l - 1. \end{cases} \qquad (6)$$

The encryption key $k_t$ at time step $t$ contains bit values $s_i^t$ at bit index $i$, and the key set $K = \{k_0, k_1, ..., k_n\}$ is then obtained by executing Eqn. (6) for $n$ iterations. We randomly select $p$ keys $K' = \{k_0', k_1', ..., k_{p-1}'\}$ from $K$ where $0 < p <= n + 1$ and sequentially perform logical exclusive OR (XOR) operations on the corresponding raw binary watermark $m_0 \in E_{\text{bin}}$ of image $I$ following

$$m_{i+1} = m_i \oplus k_i', \quad \text{for } 0 <= i < p, \qquad (7)$$

and $m_p \in M$ after $p$ sequential XOR denotes the ultimately encrypted watermark.

The encrypted watermarks are unpredictable and complex following the encryption pipeline and are used to proactively detect Deepfake materials. On the other hand, while guaranteeing confidentiality by preventing deciphering the original information from unauthorized attackers, an authorized user who owns explicit information regarding the transform pipeline and has access to the value and order of $K'$ may recover the original watermarks via XOR operations in an inverse sequence.

*3.2.3 Robustness.* To fulfill the goal of Deepfake detection, the embedded watermarks are designed to have robust manners when facing both benign and Deepfake manipulations. On the one hand, in real-life scenarios, the visual quality of images is unavoidably degraded upon posting and spreading through different protocols (e.g., on Twitter or Instagram). Consequently, the watermarks are expected to resist benign image processing operations such as compression, noising, and blurring. Therefore, we construct a benign manipulation pool $P_{\text{benign}}$ that contains the benign image processing operations to adversarially enforce the watermark robustness. On the other hand, to achieve the main objective of this study, we construct a Deepfake manipulation pool $P_{\text{deepfake}}$ that consists of various Deepfake face manipulation algorithms.

In specific, $P_{\text{benign}} = \{\text{GaussianNoise}(0, 0.1), \text{GaussianBlur}(2, 3), \text{MedianBlur}(3), \text{Jpeg}(50)\}$ and $P_{\text{deepfake}} = \{\text{SimSwap}, \text{InfoSwap}, \text{UniFace}, \text{E4S}, \text{StarGAN}, \text{StyleMask}, \text{HyperReenact}\}$ are constructed as the two pools during evaluation to validate that the watermarks are robust against a diverse of adversaries, while the model sees only Jpeg(50) and SimSwap during the training phase.

## 3.3 Auto-Encoder

We devise an auto-encoder framework and train it end-to-end for watermark embedding and recovery. The overall framework is illustrated in Figure 2. In a nutshell, we adopt mainly convolutional neural networks (CNNs) in the framework. Specifically, following a convolutional block (Conv. Block) to expand the feature dimension, repeated convolutional attention blocks (Conv. Attn. Block) are constructed to analyze the intra-correlation within the feature domain, locating the proper feature positions to hide the watermarks.

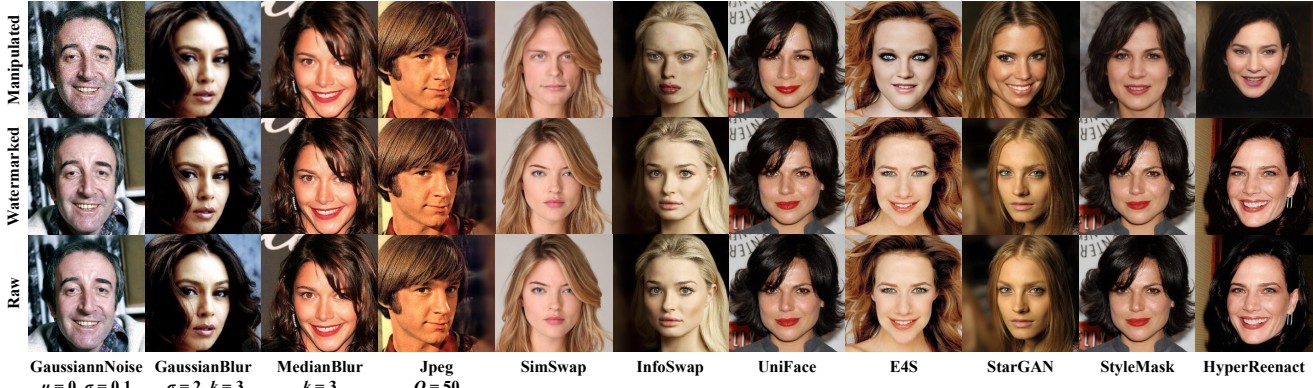

**Figure 3: Visual effects of the manipulations on the watermarked images. The raw and watermarked images are displayed in the bottom and middle rows. Manipulated outputs via different operations on the watermarked images are placed in the top row. The left four columns present results by benign manipulations and the remaining exhibit those by Deepfake manipulations.**

The attention mechanism is conducted separately and sequentially in channel-wise and spatial-wise perspectives via convolutional operations that preserve channel and spatial dimensions, respectively. The channel-wise attention is accomplished via squeeze-and-excitation networks (SENet) [12] and the spatial-wise attention is denoted by max pooling and average pooling along the channels. On the other hand, to ensure robustness, we first diffuse the watermark $m$ to match the dimension of image features, and then study the intra-correlation within the watermark features via a sequence of convolutional attention blocks. As the image and watermark features are both analyzed regarding the optimal strategies for embedding the watermarks, we concatenate the features and reconstruct the watermarked image $I_{\text{rec}}$.

Image manipulation algorithms are adopted from $P_{\text{benign}}$ and $P_{\text{deepfake}}$ and applied upon the watermarked image before passing to the decoder for watermark recovery. This setting provides representations after image manipulations so that the encoder is aware of the types of adversaries it is battling with when staying robust and the decoder can precisely recover the watermarks accordingly. As for the decoder, after a convolutional block that expands the feature channels, consecutive convolutional squeeze-and-excitation blocks (Conv. SE. Block) are applied. Unlike blocks in the encoder, the watermark features are simply refined by expanding the channels while squeezing the spatial maps without attention mechanisms. In the end, the features are flattened and linearly projected to the length of the watermarks.

In the training phase, a discriminator is established to adversarially tune the encoder for better image visual qualities. In particular, the discrimination is of similar architecture as the decoder without expanding the channel dimension, performing as a binary classifier to determine the existence of watermarks.

### 3.4 Objective Functions

Since the process of fitting the binary watermarks from facial landmarks is training-free, we focus on designing objective functions for the auto-encoder when embedding and recovering watermarks. In general, there are four objectives to be concurrently considered.

The watermarks are expected to be invisible, therefore, we apply an $L_2$ constraint to ensure reasonable visual qualities of the reconstructed watermarked images following

$$L_I = \|I_{\text{rec}} - I\|_2, \tag{8}$$

where $I$ and $I_{\text{rec}}$ are the raw and watermarked images.

At the same time, the decoder is designed to faithfully recover the embedded watermarks. Therefore, we assign an $L_2$ constraint on the decoder following

$$L_m = \|m_{\text{rec}} - m\|_2, \tag{9}$$

where $m$ and $m_{\text{rec}}$ denote the original and recovered watermarks.

While training the watermarking framework end-to-end, a discriminator $D$ that tries to distinguish raw and watermarked images is established to adversarially improve the visual qualities of the watermarked images by

$$L_{\text{adv}} = -\mathbb{E}(\log(D(I))) + \mathbb{E}(\log(1 - D(I_{\text{rec}}))). \tag{10}$$

Lastly, we reserve the benign utilization of Deepfake in the industry and employ the $L_2$ constraint to preserve the synthetic quality of Deepfake even for the target images that are watermarked,

$$L_G = \|G(I, I_s) - G(I_{\text{rec}}, I_s)\|_2, \tag{11}$$

where $G$ represents the Deepfake synthetic model and $I_s$ denotes the source image that provides synthetic information.

## 4 EXPERIMENTS

### 4.1 Implementation Details

In this study, we adopted the popular CelebA-HQ [18] and LFW [13] datasets, which both contain sufficient diversity in facial images. Specifically, we leveraged CelebA-HQ with 30,000 images and followed the official split for training, validation, and testing. Meanwhile, CelebA-HQ is employed as the data corpus to fit the transform rule $\mathbf{W}$ for watermark construction in pipeline $G_m$. LFW with 5,749 unique facial identities is used for the cross-dataset validation. While the transformation from landmarks to watermarks is training-free, we train the auto-encoder end-to-end with a learning rate of $2e - 2$ on 4 Tesla A100 GPUs.

**Table 1: Quantitative visual quality evaluation of the water-marked images. Information includes model name, image resolution, watermark length, PSNR (dB), and SSIM.**

| Model | Resolution | Length | PSNR↑ | SSIM↑ |
|---|---|---|---|---|
| HiDDeN [56] | $128 \times 128$ | 30 | 33.26 | 0.888 |
| MBRS [17] | $128 \times 128$ | 30 | 33.01 | 0.775 |
| RDA [51] | $128 \times 128$ | 100 | 43.93 | 0.975 |
| CIN [25] | $128 \times 128$ | 30 | 43.37 | 0.967 |
| ARWGAN [15] | $128 \times 128$ | 30 | 39.58 | 0.919 |
| SepMark [46] | $128 \times 128$ | 30 | 38.51 | 0.959 |
| Ours | $128 \times 128$ | 64 | **44.75** | **0.992** |
| MBRS [17] | $256 \times 256$ | 256 | 44.14 | 0.969 |
| FaceSigns [26] | $256 \times 256$ | 128 | 36.99 | 0.889 |
| SepMark [46] | $256 \times 256$ | 128 | 38.56 | 0.933 |
| Ours | $256 \times 256$ | 128 | **45.45** | **0.995** |

## 4.2 Performance Evaluation on CelebA-HQ

In this section, we validated the performance of the proposed method on CelebA-HQ [18] by evaluating the visual quality, bit-wise watermark recovery accuracy, and Deepfake detection AUC score. Contrastive methods include robust watermarking frameworks (HiDDeN [56], MBRS [17], and CIN [25]), semi-fragile proactive Deepfake watermarking frameworks (FaceSigns [26]), and robust proactive Deepfake watermarking frameworks (SepMark [46] and ARWGAN [15]). Algorithms with source code available are reproduced in all experiments.

*4.2.1 Visual Quality.* The sampled images are exhibited in Figure 3. Specifically, we visualized the original images, watermarked images, and manipulated watermarked images from bottom to top. The first four columns refer to the benign manipulations assigned with the listed parameters, and the remaining columns exhibit the effects of Deepfake manipulations, omitting the source images that provide desired identities, expressions, and head poses. It can be observed that the watermarks merely affect the visual qualities of images, and the Deepfake manipulations are regularly executed even on images with watermarks embedded.

In quantitative experiments, we computed the average peak signal-to-noise ratio (PSNR) and structural similarity index measure (SSIM) regarding the raw and watermarked images. The two metrics evaluate the level of noise and structural similarity of the watermarking framework, respectively. As Table 1 lists, our method retains outstanding visual quality in both 128 and 256 resolutions. In specific, we achieved the best performance for the two resolutions, outperforming the previous state-of-the-art algorithms. The promisingly high PSNR and SSIM values imply the imperceptible visual perturbations brought by the watermarks towards raw images. Additionally, while the early approaches generally demonstrate mere advantages, RDA [51] and CIN [25] have shown likewise reasonable visual qualities with competitive statistics at the 128 resolution, and MBRS [17] is observed to be more reliable at the 256 resolution with the second best performance.

*4.2.2 Watermark Recovery Accuracy.* We compared the similarity between the original watermark $m$ and the recovered watermark

**Table 2: Quantitative comparison on CelebA-HQ regarding the bit-wise watermark recovery accuracy of the watermarks under benign manipulations. GausNoise, GausBlur, and Med-Blur are abbreviations of Gaussian Noise, Gaussian Blur, and Median Blur for space saving.**

| Model | GausNoise | GausBlur | MedBlur | Jpeg |
|---|---|---|---|---|
| HiDDeN [56] | 51.36% | 73.04% | 82.72% | 67.84% |
| MBRS [17] | 99.60% | 99.99% | 99.99% | 99.49% |
| RDA [51] | 60.18% | 99.95% | 99.98% | 66.85% |
| CIN [25] | 86.00% | 99.99% | 97.03% | 96.86% |
| ARWGAN [15] | 53.45% | 85.22% | 96.66% | 57.42% |
| SepMark [46] | 99.25% | 99.99% | 99.99% | 99.78% |
| Ours | **99.71%** | **99.99%** | **99.99%** | **99.89%** |
| MBRS [17] | 58.31% | 72.05% | 98.06% | 99.69% |
| FaceSigns [26] | 53.61% | 98.68% | 99.86% | 82.64% |
| SepMark [46] | 99.94% | 99.99% | 99.99% | 99.99% |
| Ours | **99.99%** | **99.99%** | **99.99%** | **99.99%** |

$m_\text{rec}$, denoting the watermark recovery accuracy. Specifically, since the watermarks are binary strings with the fixed length $l$, the accuracy is derived by

$$\text{ACC}(m_\text{rec}, m) = 1 - \frac{\sum_{i=0}^{l-1} \left| m_\text{rec}^i - m^i \right|}{l}, \tag{12}$$

where $m_\text{rec}^i$ and $m^i$ refers to the bit values at index $i$ of each watermark. In this section, to verify the model robustness, watermarks are recovered after each manipulation from $P_\text{benign}$ and $P_\text{deepfake}$ is executed.

As illustrated in Table 2 and Table 3, our proposed algorithm consistently outperforms the contrastive state-of-the-art ones. In Table 2, the benign manipulations, GaussianNoise, GaussianBlur, Median-Blur, and Jpeg, are adopted for evaluation. It can be observed that, although most watermarking frameworks suffer accuracy damping due to the distortions and noises brought by the manipulations, SepMark [46] and our method generally maintain robustness in all circumstances for both resolution levels. In summary, GaussianNoise and Jpeg are observed to be the most challenging ones for all models, including ours. It is also worth noting that, despite achieving satisfactorily high statistics at the 128 resolution, results of MBRS [17] are unreliable because of the unexpectedly low visual quality in Table 1. Contrarily, while MBRS maintains reasonable visual quality at the 256 resolution, the corresponding watermark recovery accuracies in Table 2 are poor. As a result, combining the performance in Table 1 and Table 2, our approach favorably achieves state-of-the-art average watermark recovery accuracies of 99.89% and 99.99% at the 128 and 256 resolutions, respectively, against benign image manipulations while concurrently ensuring satisfactory image visual qualities.

Regarding the malicious Deepfake manipulations, watermarks are expected to stay robust so that proactive detection can be accomplished consistently. The watermark recovery accuracies at the 128 and 256 resolutions are reported in Table 3. In general, except for SepMark [46] and our approach, all remaining comparative models have derived accuracies mostly around 50%. This implies

**Table 3: Quantitative comparison on CelebA-HQ regarding the bit-wise watermark recovery accuracy of the watermarks under Deepfake manipulations. The top half refers to the 128 resolution and the bottom half refers to 256.**

|  | SimSwap [6] | InfoSwap [9] | UniFace [47] | E4S [23] | StarGAN [7] | StyleMask [3] | HyperReenact [2] | Average |
|---|---|---|---|---|---|---|---|---|
| HiDDeN [56] | 50.02% | 50.07% | 54.98% | 49.19% | 50.24% | 49.99% | 50.15% | 50.66% |
| MBRS [17] | 49.98% | 50.82% | 50.22% | 50.07% | 49.95% | 50.08% | 50.08% | 50.17% |
| RDA [51] | 50.00% | 50.01% | 71.15% | 63.03% | 47.45% | 48.94% | 56.65% | 55.32% |
| CIN [25] | 50.28% | 50.60% | 46.01% | 50.55% | 50.05% | 50.24% | 50.43% | 49.74% |
| ARWGAN [15] | 52.06% | 47.94% | 59.30% | 49.81% | 50.51% | 50.10% | 49.86% | 51.37% |
| SepMark [46] | 86.17% | 77.27% | 66.13% | 81.62% | 49.05% | 50.16% | 50.05% | 65.78% |
| Ours | **99.95%** | **97.99%** | **99.72** | **92.09%** | **73.12%** | **74.19%** | **73.53%** | **87.23%** |
| MBRS [17] | 50.00% | 50.71% | 49.98% | 50.07% | 49.95% | 50.00% | 50.07% | 50.11% |
| FaceSigns [26] | 49.74% | 50.00% | 50.59% | 49.73% | 50.51% | 49.10% | 49.28% | 49.85% |
| SepMark [46] | 92.09% | 81.49% | 57.44% | 77.32% | 50.11% | 50.06% | 50.02% | 65.50% |
| Ours | **99.98%** | **98.31%** | **94.28%** | **93.27%** | **74.66%** | **75.83%** | **74.18%** | **87.21%** |

that their watermarks are ruined by Deepfake manipulations such that the decoders are unable to recover the correct messages. As a result, although most robust watermarking frameworks are able to achieve certain levels of robustness regarding some of the benign image manipulations, they are unfavorably unsatisfied against Deepfake manipulations. In addition, FaceSigns [26], although designed as a semi-fragile watermarking framework that is supposed to be vulnerable and have low accuracies when facing Deepfake manipulations, fails to stay robust against benign image manipulations. Meanwhile, although trained against SimSwap, SepMark still suffers considerable watermark recovery errors with 86.17% and 92.09% accuracies for the two resolutions. Furthermore, strong fluctuations can be observed under the cross-manipulation scenario such that unseen Deepfake manipulations have caused significant challenges to SepMark by destroying the underlying watermarks.

In conclusion, our proposed algorithm maintains the best robustness for benign and Deepfake manipulations and achieves 87.23% and 87.21% average accuracies in Table 3 for the 128 and 256 resolutions, respectively. At the same time, despite being trained against SimSwap [6] solely regarding Deepfake manipulations, our model demonstrates state-of-the-art cross-manipulation performance when tested with other Deepfake manipulations. Additionally, while statistics for face swapping models are all above 90%, those for face reenactment models are only around 75%. This is possibly caused by wiping out the background information in face reenactment results, making it more difficult to recover the original watermarks.

4.2.3  *Deepfake Detection.* In this study, the ultimate goal of maintaining watermark robustness is to ensure the reliability of watermarks and enforce proactive Deepfake detection accordingly. Particularly, for a watermarked image $I_{\text{rec}}$ with embedded landmark perceptual watermark $m_{\text{raw}}$ derived from the raw image $I_{\text{raw}}$ following the pipeline $G_m$, falsification is addressed based on the recovered watermark $m_{\text{rec}}$ and landmark perceptual watermark $m_{\text{sus}}$ of a suspect image $I_{\text{sus}}$. If $I_{\text{sus}}$ is derived via benign manipulations, the structural content is not modified and the similarity between $I_{\text{sus}}$ and $m_{\text{rec}}$ is, therefore, expectedly high. On the other hand, if $I_{\text{sus}}$ is a Deepfake image, due to structural changes in the

image content, $I_{\text{sus}}$ and $m_{\text{rec}}$ are dissimilar. Based on this rule, we conducted Deepfake detection on the seven Deepfake manipulation algorithms and computed the AUC scores by concurrently introducing benign image manipulations upon the raw images to produce real samples.

In Table 4, we compared the detection performance with four popular and state-of-the-art passive Deepfake detectors[2], namely, Xception [31], SBIs [32], RECCE [4], and CADDM [8]. While the four passive detectors have demonstrated superior detection ability in lab-controlled scenarios on the FaceForensics++ [31] dataset, as listed in the last row of Table 4, they are generally fooled by the up-to-date synthetic algorithms with as low as less than 40% for AUC scores. Specifically, besides the lower resolution level bringing more difficulty with indistinguishable underlying artifacts and noises, the hyper-realistic output images by E4S, StarGAN, and StyleMask have limited the AUC scores of detectors all below 80%. On the other hand, SBIs and CADDM are observed to be the most powerful passive detectors by closely approaching 90% for AUC scores on some Deepfake manipulations.

As for our proposed proactive approach, relying on the outstanding watermark recovery accuracy for all benign and Deepfake manipulations, the extracted watermarks by the decoder are faithfully similar to the originally embedded ones. Therefore, comparing $I_{\text{sus}}$ and $m_{\text{rec}}$ leads to reliable Deepfake detection results. As reported in Table 4, our approach demonstrates superior detection ability with AUC scores all above 95% regardless of manipulation algorithms. In other words, the similarities between $I_{\text{sus}}$ and $m_{\text{rec}}$ for the benign manipulations are expectedly higher than those for the Deepfake manipulations. To conclude, our approach is proved to have the best performance of 98.39% and 98.55% for AUC scores even with a mixture of all seven Deepfake manipulations.

## 4.3  Cross-Dataset Evaluation

Under the cross-dataset setting, we further evaluated the generalizability of our proposed LampMark on unseen datasets. In particular, we adopted the LFW dataset and conducted experiments at the 128

---

[2]The proactive approaches are excluded since the unsatisfactory watermark recovery accuracies make the corresponding detection unconvincing. In addition, since their watermarks are semantically void, the pipeline for Deepfake detection is incomplete.

**Table 4: Deepfake detection performance in AUC scores against different face manipulation algorithms on CelebA-HQ at 128 and 256 resolutions. The row of 'Mixed' evaluates the detection performance on a mixed testing set of all seven algorithms.**

| | Xception [48] | | SBIs [32] | | RECCE [4] | | CADDM [8] | | Ours | |
|---|---|---|---|---|---|---|---|---|---|---|
| Resolution | 128 | 256 | 128 | 256 | 128 | 256 | 128 | 256 | 128 | 256 |
| SimSwap [6] | 39.37% | 71.15% | 75.30% | 88.94% | 60.37% | 69.01% | 55.91% | 87.66% | **97.80%** | **99.01%** |
| InfoSwap [9] | 60.82% | 65.50% | 85.11% | 80.50% | 55.51% | 52.13% | 48.29% | 61.39% | **98.59%** | **99.18%** |
| UniFace [47] | 71.79% | 70.34% | 72.45% | 79.41% | 61.58% | 67.35% | 82.16% | 82.73% | **96.76%** | **97.03%** |
| E4S [23] | 43.40% | 53.70% | 63.63% | 61.05% | 60.88% | 47.19% | 64.93% | 73.13% | **98.99%** | **99.10%** |
| StarGAN [7] | 37.14% | 40.30% | 48.98% | 65.86% | 35.82% | 41.55% | 37.41% | 44.34% | **98.96%** | **99.32%** |
| StyleMask [3] | 29.41% | 40.23% | 38.45% | 48.45% | 31.08% | 23.87% | 34.87% | 39.73% | **98.62%** | **98.98%** |
| HyperReenact [2] | 38.96% | 76.27% | 52.36% | 53.35% | 82.23% | 78.23% | 35.87% | 42.87% | **98.87%** | **99.02%** |
| Mixed | 41.28% | 41.42% | 60.39% | 68.62% | 54.09% | 52.51% | 52.04% | 59.84% | **98.39%** | **98.55%** |
| FF++ [31] | 97.60% | | 90.50% | | 96.81% | | 95.57% | | – | |

**Table 5: Quantitative experiments on LFW at the 128 resolution for visual quality and bit-wise watermark recovery accuracy under benign and Deepfake manipulations.**

| | Hidden [56] | MBRS [17] | RDA [51] | CIN [25] | ARWGAN [15] | SepMark [46] | Ours |
|---|---|---|---|---|---|---|---|
| GaussianNoise | 51.30% | 99.80% | 60.83% | 87.54% | 53.11% | 81.90% | **99.90%** |
| GaussianBlur | 72.89% | 99.99% | 99.88% | 99.99% | 86.39% | 91.97% | **99.99%** |
| MedianBlur | 82.35% | 99.99% | 99.98% | 99.98% | 96.93% | 89.72% | **99.99%** |
| Jpeg | 56.17% | 99.42% | 66.77% | 97.06% | 70.86% | 83.42% | **99.95%** |
| Average | 65.68% | 99.80% | 81.87% | 96.14% | 76.82% | 86.75% | **99.95%** |
| SimSwap [6] | 49.96% | 49.92% | 50.04% | 50.25% | 50.00% | 59.72% | **99.58%** |
| InfoSwap [9] | 50.06% | 50.04% | 50.10% | 50.95% | 50.86% | 76.17% | **91.59%** |
| UniFace [47] | 53.74% | 49.78% | 70.37% | 49.92% | 59.16% | 63.10% | **97.58%** |
| E4S [23] | 49.34% | 50.08% | 67.78% | 50.22% | 48.64% | 88.14% | **94.53%** |
| StarGAN [7] | 50.09% | 49.97% | 48.95% | 50.25% | 49.83% | 50.05% | **66.15%** |
| StyleMask [3] | 50.08% | 49.97% | 49.59% | 50.14% | 40.83% | 49.07% | **70.42%** |
| HyperReenact [2] | 50.13% | 49.91% | 56.41% | 50.21% | 49.83% | 49.02% | **66.28%** |
| Average | 50.49% | 49.95% | 56.13% | 50.28% | 40.88% | 62.18% | **83.73%** |
| PSNR↑ | 33.27 | 32.78 | 42.71 | 42.89 | 39.94 | 37.03 | **43.14** |
| SSIM↑ | 0.883 | 0.761 | 0.959 | 0.982 | 0.926 | 0.947 | **0.983** |

resolution following the same pipeline as Section 4.2 illustrates. Experimental results are reported in Table 5. It can be seen that the watermarking frameworks generally perform similarly to those on CelebA-HQ. Specifically, besides the visual qualities of RDA [51], CIN [25], and ours being generally acceptable, GaussianNoise and Jpeg are still the most challenging benign image manipulations that have caused the most trouble in watermark recovery.

Meanwhile, the watermarks behave in like manners as in Table 2 when evaluated against Deepfake manipulations such that most of them become fragile when being manipulated by Deepfake. As a result, although slightly suffers the cross-dataset challenge when tested against the face reenactment models, the proposed method consistently reaches state-of-the-art performance with an 89.73% recovery accuracy on average for all manipulations. Furthermore, SepMark, besides consistently demonstrating fragile characteristics when facing unseen manipulations, encounters similar performance damping as ours under the cross-dataset setting.

## 5 CONCLUSION

In this work, we exploit the structure-sensitive characteristic of Deepfake manipulations and present a proactive Deepfake detection approach, LampMark, that relies on landmark perceptual watermarks. We propose a training-free pipeline to transform facial landmarks into binary watermarks, securely protected by a sophisticated cellular automaton encryption system. Then, an end-to-end auto-encoder architecture is trained to embed and extract watermarks robustly. Extensive experiments demonstrate the state-of-the-art performance of our method in watermark recovery and Deepfake detection. Lastly, despite successfully stepping forward with the innovative idea, the existing performance gap from perfect statistics points out the future direction of our research to continuously advance the model generalizability, especially for the cross-manipulation watermark recovery ability against face reenactment manipulations.

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
