# OpenReview forum: "Proactive Deepfake Detection via Training-Free Landmark Perceptual Watermarks"
_acmmm.org/ACMMM/2024/Conference — MM2024 Poster_

### Official Review · Reviewer_sUpQ · 2024-05-18

**Rating:** 5
**Confidence:** 4

**Summary:**

This paper introduces a novel proactive Deepfake detection approach via training-free landmark perceptual watermarks. By analyzing the structure-sensitive characteristics and assigning landmark perceptual semantics to the robust watermarks, the proposed LampMark accomplishes the proactive detection pipeline. The authors also devise an encryption pipeline via a cellular automaton system to ensure watermark confidentiality. The paper reports promising watermark recovery and Deepfake detection results compared to the SOTA methods.

**Strengths:**

1. The idea of the landmark perceptual watermark is novel, and it assigns semantics to the watermarks in a training-free manner, thus enhancing the explainability of Deepfake detection results at a low cost.
2. This work innovatively accomplishes the entire pipeline of proactive Deepfake detection without knowing the groundtruth in advance, which is barely achieved in previous studies.
3. This paper provides good statistics with sufficient experiments for in-dataset, cross-dataset, and cross-manipulation scenarios, demonstrating reliable model performance against various SOTA algorithms.

**Limitations:**

1. There is an inconsistency of numerical statistics on page 2 in section 1 when reporting the average watermark recovery accuracies, 91.83% and 91.86%, compared to Table 3. Please double-check and clarify this in rebuttal.
2. There are other commonly seen image processing operations such as brightness, contrast, saturation, and hue. Since they have been discussed in recent papers, how robust is LampMark when facing these operations?
3. Typo issue to be fixed: A missing percentage notation in Table 3 at performance for UniFace.

**Suitability:**

3

---

### Official Review · Reviewer_ZPyT · 2024-05-24

**Rating:** 3
**Confidence:** 3

**Summary:**

The security and privacy issues have always been the key issue for the application of deepfake technique, especially in face manipulation. This paper proposes a proactive Deepfake detection approach based on watermarking scheme, this can not only perform forgery detection when the manipulation is maliciously, but also can preserve the quality of watermarked image and avoid hindering the benign application of deepfake technology. The main contribution is to introduce a landmark perceptual watermark transformation pipeline based on the structure-sensitive characteristics of face manipulation, which can satisfy the confidentiality, discrimination and robustness simultaneously. Evaluation experiments comprehensively cover the in-dataset, cross-dataset, and cross-manipulation settings, demonstrating the superior detection performance of the proposed method.

**Strengths:**

- This paper contributes a proactive deepfake detection framework based on watermarking scheme, which innovatively exploit the structure-sensitive characteristics of face manipulations, and combine the concept of cellular automata to ensure confidentiality.
- Datasets and baseline methods are comprehensive, and the results can fully support the proposed framework LampMark.

**Limitations:**

- The biggest limitation of LampMark framework is requiring the original face image in the inference detection, that is, it is a non-blind watermarking scheme, which I think is limited in real-world scenario.
- In the training stage, the decoder receives both benign and fake images, and if the forged image is also embedded in the landmark perceptual watermark, whether it will interfere with the recovery of the original image watermark.
- I wonder to know how to constrain and optimize $m_{rec}$, as described in Section 3.1, the objective functions seems lack of the constraint on $<m_{benigh},m_{rec}>$  and $<m_{fake},m_{rec}>$. In addition, if training end-to-end, there should be a total objective function and the  a description of weights for each part.
- I am curious about why the model sees only Jpeg(50) and SimSwap during the training phase, but it shows good robustness to more types of distortion at testing phase, please give a detailed explanation.
- The presentation of this paper needs to improve, and there are some notations are not uniform. Such as, the $m$ in Eq(9) and  $m_{raw}$ in Section 3.1 seems both refer to the landmark perceptual watermark of the original image, and $I$ in Eq(8) seems to be $I_{raw}$, please take care to make a uniform representation. Additionally, there is a lack of notation for the  input image of $P_{deepfake}$ in Figure 2 and there are still some reference citations problems, the authors need carefully check the manuscript.

**Suitability:**

3

---

### Official Review · Reviewer_bJzC · 2024-05-24

**Rating:** 5
**Confidence:** 2

**Summary:**

This work addresses the problem of deepfake detection. The work introduces a novel training-free landmark perceptual watermark, which is used to detect two types of deepfakes: face swapping and face re-enactment. The experimental results show that both watermark extraction and deepfake detection are achieved with very good performance results.

**Strengths:**

- The writing is clear, easy to understand and the narrative is logical.
- This is an issue that has been on the minds of the community since depth modelling became active.
- The methodology and evaluation process is very detailed and solid.

**Limitations:**

I have to be honest and say that I am not very familiar with the subfield of depth forgery, and I have carefully studied the related work and the methods and experimental procedures presented in this entry. It may be difficult for me to make particularly nuanced and focused comments. One suggestion I have is to perhaps include more discussion in the experimental results evaluation section, e.g. why does your method achieve such good experimental results? Which steps played a key role?

**Suitability:**

2

---

### Official Review · Reviewer_JREk · 2024-06-01

**Rating:** 5
**Confidence:** 2

**Summary:**

In this paper, the authors proposed a proactive deepfake detection method by using a trainning-free landmark perceptual watermark. The end-to-end watermarking framework is proposed to improve the robustness of the extraction. The main contribution lies in the transformation from the facial landmarks to the perceptual watermarks. Overall, the manuscript is well organized and the experimental comparisons are sufficient.

**Strengths:**

The algorithm is designed in a sophisticated manner. Each step and modul are given with reasonable motivations.

**Limitations:**

The improvement over the SOTA method is quite limited. Is it valuable to put forward the investigation on this specific problem, as the current methods have shown a very good result?

**Suitability:**

3

---

### Official Review · Reviewer_8Tma · 2024-06-03

**Rating:** 4
**Confidence:** 2

**Summary:**

The paper titled "Proactive Deepfake Detection via Training-Free Landmark Perceptual Watermarks" introduces a novel approach to Deepfake detection. The authors propose a proactive method named LampMark that leverages facial landmarks to create perceptual watermarks. These watermarks are binary representations generated from facial structures, which are then embedded into images to protect them against potential manipulations. The system is designed to be robust against both benign image processing operations and Deepfake manipulations. It employs an auto-encoder architecture for watermark embedding and recovery, ensuring the watermarks remain intact even after various manipulations. The paper claims state-of-the-art watermark recovery and Deepfake detection performance across different scenarios.

**Strengths:**

1.	The paper presents a unique solution to Deepfake detection by using perceptual watermarks based on facial landmarks, which differs from traditional passive detection methods.
2.	LampMark's watermark generation process does not require training, which can save significant time and resources compared to machine learning-based methods.
3.	The paper reports high-performance metrics, with excellent watermark recovery accuracies and AUC scores for Deepfake detection, outperforming existing state-of-the-art methods.

**Limitations:**

1.	LampMark utilizes auto-encoders, yet the experimental section lacks clarity on the auto-encoders' configuration. This ambiguity could hinder future researchers' ability to replicate the system and assess the robustness of its architecture.
2.	Lack of ablation experiments: LampMark framework consists of encoder and decoder but does not test the performance of the different auto-encoder configurations.
3.	The descriptive details of the article need to be checked. Such as line 458, ‘we adopt mainly…’.

**Suitability:**

3

---

### Meta-Review · Area_Chair_6qQe · 2024-07-01

**Recommendation:** Accept (Poster)
**Confidence:** 5

**Metareview:**

The reviewers generally are satisfied with the rebuttal, and recognize the contribution of proposed method along with the convincing evaluations.
Basically, the contributions of this manuscript include (1) a new solution to deepfake detection based on perceptual watermarking; (2) LampMark is training-free; (3) experimental evaluation is coprehensive.
Based on the above concern, it is recommended to accept this manuscript.